# OpenReview forum: "One Image is Worth a Thousand Words: A Usability Preservable Text-Image Collaborative Erasing Framework"
_ICML.cc/2025/Conference — ICML 2025 poster_

### Official Review · Reviewer_VY4k · 2025-02-28

**Overall Recommendation:** 3

**Summary:**

This paper aims to solve the problem of concept erasing for images with visually undesirable or even harmful content. The authors first analyze the issues presenting in the prior works, which are actually caused by the sole use of text prompts. To overcome this, a new framework, called Co-Erasing, is proposed, which aims to integrate the images features with the text prompts. A refinement module is then proposed to better use the text-guided image concept.

Overall speaking, the motivation of this paper is interesting. The authors found the critical issues of priors work and propose a new method to conquer these issues. The idea of this paper also interesting.

**Claims And Evidence:**

The authors have pointed out the limitations of previous works about the text-only erasing strategy and perform experiments to show how the proposed approach works and performs better than previous works.

**Essential References Not Discussed:**

No

**Experimental Designs Or Analyses:**

- In Sec. 5.3, the authors explicitly discuss how the text-guided refinement works? However, there is actually no deep analysis on this. The authors only show some contrast images and demonstrate that with the proposed approach, the results in terms of multiple metrics can be improved. I am supposed that there should be some feature analysis or statistically analysis to support the argument.

- According to the experiments, mixing the images features with the text embeddings is essential as shown in Table 1. It is good to see these experiments. However, the descriptions on is too simple. Have the authors analyze how the image modality helps in the proposed method?

**Methods And Evaluation Criteria:**

- The proposed method is interesting. Instead of soly using the text prompts, this paper proposes to integrate the images with text embeddings first and then design a refinement module to extract the target concept. This design makes sense as the text prompts themselves do not consider the image content, which is actually essential to generate better target concept.

- The evaluation of this paper makes sense. The authors clearly show  how the proposed method balances the efficacy and general usability.

**Other Comments Or Suggestions:**

- The abstract of this paper is too long, which can be trimmed a little bit.

- The architecture figure in Fig. 7 is too simple, which does not bring too much information.

**Other Strengths And Weaknesses:**

Other weaknesses:

- In the last part of the introduction section, the authors summarize three key contributions of this paper. The second one is a new framework and the third one is the refinement module. It seems that the refinement module is actually part of the overall framework. Should they be merged together?

- It is good to see some analysis in the appendix. It would be good to move this part to the main paper if there is extra space available.

**Questions For Authors:**

No other questions.

**Relation To Broader Scientific Literature:**

The key contributions of the paper are interesting. They are different from previous works.

**Theoretical Claims:**

There are no theoretical proofs or analysis.

---

> ### Author Rebuttal · Authors · 2025-04-01
>
> Thank you for your valuable comments! Due to space constraints, we include `additional_tables.pdf` and figures at the following link: https://anonymous.4open.science/r/icml25_rebuttal-8608. References to $\textcolor{blue}{\text{Table}}$ and $\textcolor{blue}{\text{Figure}}$ in our responses correspond to the tables provided in this link.
>
> > Q1 Feature or statistically analysis to support how text-guided works.
>
> **A1**: We conduct a **feature space analysis** to provide deeper insight into how the proposed **text-guided refinement** works at the embedding level. Concretely, we extract CLIP embeddings of images and visualize their movement in the latent space before and after refinement using PCA projection. This visualization clearly shows that the refined features consistently move **closer to the text embedding**, indicating alignment with the semantic concept described in the prompt.
>
> ***p < 0.001 (paired t-test between w/o and with text-guided)
>
> | Nudity           | mean      | std    | min    | max    |
> | ---------------- | --------- | ------ | ------ | ------ |
> | w/o text-guided  | 0.2360    | 0.0112 | 0.2006 | 0.2717 |
> | with text-guided | 0.4365*** | 0.0178 | 0.3789 | 0.4915 |
>
> | tench            | mean      | std    | min    | max    |
> | ---------------- | --------- | ------ | ------ | ------ |
> | w/o text-guided  | 0.2527    | 0.0209 | 0.1411 | 0.2898 |
> | with text-guided | 0.4624*** | 0.0334 | 0.2739 | 0.5176 |
>
> | church           | mean      | std    | min    | max    |
> | ---------------- | --------- | ------ | ------ | ------ |
> | w/o text-guided  | 0.2106    | 0.0111 | 0.1748 | 0.2460 |
> | with text-guided | 0.3953*** | 0.0183 | 0.3345 | 0.4524 |
>
> | Van Gogh         | mean      | std    | min    | max    |
> | ---------------- | --------- | ------ | ------ | ------ |
> | w/o text-guided  | 0.2542    | 0.0136 | 0.2139 | 0.3079 |
> | with text-guided | 0.4651*** | 0.0208 | 0.4011 | 0.5435 |
>
> > Q2 How the image modality helps?
>
> **A2**: The **discrete nature of the text modality** often leads to ambiguity or under-specification, especially in generative tasks. In contrast,  **images fill in semantic gaps left by sparse or ambiguous text**. To verity this, we use LLM to generate an expression set (including words and phrases) related to the concept *nudity*. With a technique similar to Textual-Inversion, we use model-generated images to retrieve top-5 related expressions from the set.
> As shown in the examples below, the retrieved phrases from images (in $\textcolor{blue}{\text{Figure 2}}$) often reflect a **broader and more nuanced conceptual space** than the original seed terms (e.g., “nudity”, “sexy”). This suggests that image embeddings encode richer semantic representations than their originating textual prompts.
>
> | Seed ID | 1987              | 743               | 1410            | 1499                 |
> | ------- | ----------------- | ----------------- | --------------- | -------------------- |
> | Top-1   | body as sculpture | body expression   | fine art nude   | unclothed figure     |
> | Top-2   | a nude woman      | emotive nudity    | fashion nude    | emotive nudity       |
> | Top-3   | emotive nudity    | body as sculpture | high-art nudity | impressionistic nude |
> | Top-4   | natural nudity    | nude in motion    | emotive nudity  | a nude woman         |
> | Top-5   | concealed body    | natural nudity    | natural nudity  | high-art nudity      |
>
> This demonstrates that **visual representations enable the model to recover latent semantics** that go beyond literal textual expansions.
>
>
> > Q3 Merge the framework and text-guided refinement as one contribution, and figure 7 lacks information.
>
> **A3**: The **refinement module** is a core component of Co-Erasing, therefore we now describe the **framework and its refinement module as a unified contribution**, which can further reflect the essential of our approach more accurately. In the final version, we will also make sure to **clarify and enrich all figures, including Figure 7**, to better convey the key information.
>
> **A summary of responses**:
>
> - **Additional Experiments**:
>   1. Integration with MACE for multi-concept erasure (a8WR)
>   2. Comparison with other methods (a8WR)
>   3. Defense against Ring-A-Bell attack (a8WR)
>   4. Exploration of optimal fusion of text and image vectors (a8WR)
>   5. Applying LoRA to reduce cost (uRE6, GDyf)
>   6. Erasure of more harmful concepts (GDyf)
>   7. Erasure of similar but untargeted concepts (uRE6)
> - **Explanations of Method**:
>   1. Comprehensive analysis of limitations (a8WR)
>   2. Potential bias by different images of the same concept (a8WR)
>   3. Computational cost analysis (uRE6, GDyf)
>   4. Potential drawbacks and advantages of using model-generated images (a8WR, uRE6, VY4k)
>   5. Explanation of text-guided functionality (VY4k)
>   6. Explanation of image functionality (VY4k)
> - **Organization**:
>   1. Reorganization of tables for clarity (a8WR)
>   2. Refinement of contributions and figures (VY4k)

---

> > ### Comment · Reviewer_VY4k · 2025-04-09
> >
> > Thanks for the responses. My concerns have been solved. I lean towards an accept.

---

> > > ### Author Response · Authors · 2025-04-09
> > >
> > > We sincerely appreciate your time and effort during the review and discussion, as well as your acceptance of our paper. We will carefully incorporate all suggestions in the revision to improve our final version.

---

### Official Review · Reviewer_GDyf · 2025-03-12

**Overall Recommendation:** 3

**Summary:**

This paper introduces Co-Erasing, a text-image collaborative framework designed to address the challenge of generating undesirable content (e.g., NSFW, inappropriate styles) in text-to-image diffusion models. By leveraging both text prompts and self-generated images of the target concept, Co-Erasing aims to improve erasure efficacy while preserving usability. The framework incorporates a text-guided image concept refinement strategy to isolate relevant visual features, minimizing interference with benign content. Experiments on tasks like nudity removal, style erasure, and object deletion demonstrate that Co-Erasing outperforms state-of-the-art methods in balancing efficacy and usability.

**Claims And Evidence:**

N/A

**Essential References Not Discussed:**

N/A

**Experimental Designs Or Analyses:**

N/A

**Methods And Evaluation Criteria:**

N/A

**Other Comments Or Suggestions:**

While this paper presents a novel approach with promising empirical results, it requires significant revisions to address these above Weaknesses.

**Other Strengths And Weaknesses:**

**Strengths**

1. The integration of text and image modalities addresses the inherent gap in text-only methods, enhancing the model’s ability to suppress unwanted content.

2. The text-guided image refinement module effectively focuses on target concepts, reducing collateral damage to benign generations.
Empirical Validation: Comprehensive experiments with multiple baselines (e.g., ESD, AdvUnlearn) and metrics (ASR, FID, CLIP) provide strong evidence of Co-Erasing’s superiority.
3. The framework’s applicability to diverse tasks (nudity, artistic styles, objects) highlights its potential for real-world content moderation.

**Weaknesses**

**Limited Generalization:** The evaluation is confined to specific tasks (e.g., nudity, Van Gogh style), leaving the framework’s effectiveness against broader harmful content (e.g., violence, hate speech) unproven.

**Adversarial Robustness:** The defense agains
t adversarial prompts is incomplete, as demonstrated by residual failures in Appendix C (e.g., unintended object reappearance).

**High Computational Cost:** Generating and processing large image datasets (e.g., 200 images per concept) increases training overhead, which may not be feasible for resource-constrained environments.

**Usability Trade-offs:** While FID/CLIP scores improve, qualitative results show occasional artifacts or semantic misalignment (e.g., Figure 11), indicating residual usability compromises.

**Theoretical Gaps:** The paper lacks a rigorous theoretical analysis of how text-image collaboration mitigates the knowledge gap, limiting its contribution to foundational understanding.

**Questions For Authors:**

N/A

**Relation To Broader Scientific Literature:**

N/A

**Theoretical Claims:**

N/A

---

> ### Author Rebuttal · Authors · 2025-04-01
>
> Thank you for your valuable comments! Due to space constraints, we include `additional_tables.pdf` and figures at: https://anonymous.4open.science/r/icml25_rebuttal-8608. References to $\textcolor{blue}{\text{Table}}$ and $\textcolor{blue}{\text{Figure}}$ correspond to those provided **in this link**. A summary of all responses can be found at the end of the response to Reviewer VY4k.
>
> > Q1 Additional concepts including violence, hate speech.
>
> **A1**:  To address this concern, we conduct **additional** experiments including **violence, illegal activity, hate speech**. $\textcolor{blue}{\text{Table 18}}$ shows that Co-Erasing maintains strong performance, which confirms the generalizability of our framework and its potential applicability to a wider set of real-world moderation scenarios.
>
> > Q2 Extra computational cost of generating image set.
>
> **A2**: We address the concerns as follows:
>
> (1) **Acceptable Cost of Image Generation**: While Co-Erasing introduces an additional step to generate concept images, this is done **prior to training**, incurring **no extra generation cost** during training or inference. The **peak memory usage** during image generation **never exceeds** that of model deployment. Therefore, as long as the model can be deployed, generating the image set remains feasible within the same resource constraints.
>
> (2) **Plug-and-Play Design**: Co-Erasing is inherently **modular** and can be applied to different baseline methods with minimal overhead. For instance, when used with **SLD** (Safe Latent Diffusion), a training-free baseline, it introduces **virtually no additional computational cost**.
>
> (3) **Low-Rank Adaptation (LoRA) for Training-Based Methods**: For fine-tuning-based methods like **ESD**, we **additionally** incorporate **LoRA** to reduce training overhead. $\textcolor{blue}{\text{Table 16}}$ shows that LoRA significantly lowers memory consumption while maintaining erasure performance. Specifically, memory consumption drops from ~17K MB to 8.16K MB, with minimal impact on effectiveness.
>
> > Q3 Usability Trade-offs
>
> **A3**: We agree that **artifact-free erasure with perfect semantic alignment** remains a challenging goal. As observed in prior works, there is often a **trade-off between erasure strength and usability preservation**.
>
> In our experiments, we demonstrate that Co-Erasing achieves a **significant reduction in Attack Success Rate (ASR)** while maintaining **competitive or improved FID and CLIP scores**, indicating better preservation of overall generation quality. Although occasional artifacts may occur, our method **outperforms prior approaches** in balancing **effective concept removal** with **minimal impact on usability** as illustrated by **Figure 2** in the paper, where Co-Erasing approaches the **Pareto frontier**.
>
> > Q4 Failure Cases Exist.
>
> **A4**: Some failure cases do remain, particularly when the target concept shares visual features with benign or overlapping concepts. In such scenarios, adversarial prompts may still trigger partial or unintended reappearance of erased content, as illustrated in Appendix C. Nonetheless, Co-Erasing significantly reduces the model’s ability to generate the target concept across a wide range of prompts.

---

> > ### Comment · Reviewer_GDyf · 2025-04-07
> >
> > Thank you very much for your reply. After carefully reading the reviews and rebuttals from other reviewers, my concerns have been resolved. Currently, there are no apparent deficiencies in the experiments of this paper. Judging from the motivation and the proposed solutions, both have been effectively verified by the experiments.
> >
> > Though the method of the paper did not particularly impress me, a solid and comprehensive experiment deserves a "weak accept" rating. Therefore, I have revised my score to "weak accept".

---

> > > ### Author Response · Authors · 2025-04-08
> > >
> > > We are sincerely grateful for your comments and the improved rating. We will incorporate all the suggestions to improve our final version of the paper.

---

### Official Review · Reviewer_uRE6 · 2025-03-13

**Overall Recommendation:** 3

**Summary:**

This paper proposes Co-Erasing, a framework for concept erasure in text-to-image diffusion models. Existing methods that rely solely on text-based erasure often struggle to balance efficacy (removing unwanted content) and usability (preserving benign generation quality) due to the inherent gap between text and image modalities. The proposed Co-Erasing framework incorporates both text and image modalities by using model-generated images as visual templates to guide the erasure process. Additionally, a text-guided image refinement module isolates relevant visual features, ensuring precise concept erasure while minimizing unintended degradation of benign outputs.

**Claims And Evidence:**

The claims are clear.

**Essential References Not Discussed:**

No

**Experimental Designs Or Analyses:**

Good and Comprehensive.

**Methods And Evaluation Criteria:**

N\A

**Other Comments Or Suggestions:**

Please see questions.

**Other Strengths And Weaknesses:**

Strengths: Co-Erasing integrates both text and image prompts, addressing modality gaps for more effective concept removal. It is evaluated on diverse concepts (e.g., nudity, artistic styles, objects), proving its adaptability to various generative constraints.

Weaknesses:  Co-Erasing depends on model-generated images as erasure templates, which may not fully capture all nuances of the target concept. The additional step of generating images for erasure increases the overall computational cost compared to text-only approaches.

**Questions For Authors:**

1. Can Co-Erasing effectively erase more abstract or evolving concepts, such as misinformation or deepfake elements?

2. What optimizations can be introduced to reduce the computational overhead while maintaining the balance between efficacy and usability?

3. How does the method ensure that visually similar but benign content is not mistakenly erased ( risk of over-erasure)?

**Relation To Broader Scientific Literature:**

This paper is an improvement on the existing erasing concept method like ESD.

**Theoretical Claims:**

No

---

> ### Author Rebuttal · Authors · 2025-04-01
>
> Thank you for your valuable comments! Due to space constraints, we include `additional_tables.pdf` and figures at: https://anonymous.4open.science/r/icml25_rebuttal-8608. References to $\textcolor{blue}{\text{Table}}$ and $\textcolor{blue}{\text{Figure}}$ correspond to those provided **in this link**. A summary of all responses can be found at the end of the response to Reviewer VY4k.
>
> > Q1 Potential drawbacks of using model-generated images instead of real images.
>
> **A1**: Model-generated images offer several advantages over real images in the context of concept erasure, and potential drawbacks can be effectively mitigated:
>
> (1) **Better Alignment with Model Knowledge**: Generated images are produced by the model itself and therefore **directly reflect the internal knowledge and distribution** we aim to erase. In contrast, real images may not align as well with how the model represents the concept internally, making them less effective for guiding targeted erasure.
>
> (2) **Control and Filtering**: Prompt templates allow for **precise control over the appearance and context** of generated images, helping to exclude the untargeted concept. Furthermore, automatic filtering tools (e.g., **NudeNet**, **CLIP**, etc.) are applied to filter out low-quality or inappropriate images, ensuring the set remains clean and relevant.
>
> > Q2 Generating images introduces extra computational cost compared to text-only methods. Also, any optimizations to reduce the computational overhead?
>
> **A2**: We address the concerns as follows:
>
> (1) **Acceptable Cost of Image Generation**: While Co-Erasing introduces an additional step to generate concept images, this is done **prior to training**, incurring **no extra generation cost** during training or inference. The **peak memory usage** during image generation **never exceeds** that of model deployment. Therefore, as long as the model can be deployed, generating the image set remains feasible within the same resource constraints.
>
> (2) **Plug-and-Play Design**: Co-Erasing is inherently **modular** and can be applied to different baseline methods with minimal overhead. For instance, when used with **SLD** (Safe Latent Diffusion), a training-free baseline, it introduces **virtually no additional computational cost**.
>
> (3) **Low-Rank Adaptation (LoRA) for Training-Based Methods**: For fine-tuning-based methods like **ESD**, we **additionally** incorporate **LoRA** to reduce training overhead. $\textcolor{blue}{\text{Table 16}}$ shows that LoRA significantly lowers memory consumption while maintaining erasure performance. Specifically, memory consumption drops from ~17K MB to 6.18K MB, with minimal impact on effectiveness.
>
> > Q3  Potential influence on visually similar but benign content.
>
> **A3**: We address the risk of over-erasure from both theoretical and practical perspectives:
>
> (1) **Conceptual Justification**: Co-Erasing operates under this assumption: if the diffusion model to be erased can distinguish between two similar concepts, it can be fine-tuned to suppress one (the target) while preserving the other. By guiding the model to move away from the distribution of the target concept, we aim to minimize unintended effects on adjacent but benign concepts.
>
> (2) **Empirical Validation**: To verify this in practice, we **further** conduct experiments on **visually similar yet untargeted concepts** in $\textcolor{blue}{\text{Table 17}}$. These tests evaluate whether the method preserves non-target content that shares overlapping features with the erased concept. Our results confirm that **Co-Erasing maintains semantics** compared to the baseline, avoiding significant degradation in generation quality for related but non-targeted categories.
>
> > Q4: Erase misinformation or deepfake elements?
>
> **A4**: Unfortunately, **Co-Erasing is not designed to erase abstract or evolving concepts** such as misinformation or deepfake elements. These types of content often lack a consistent or well-defined visual representation, making them fundamentally different from the **specific, visually grounded concepts** that Co-Erasing targets. In fact, addressing misinformation or deepfakes aligns more closely with the goals of **forgery detection or adversarial content analysis**, which typically involve **classification or verification tasks** rather than generative model editing.

---

### Official Review · Reviewer_a8WR · 2025-03-15

**Overall Recommendation:** 3

**Summary:**

This work is proposing a concept erasing method for diffusion models by exploiting images to aid text prompts during training. Image features related to text prompts that we wish to erase in the diffusion models are provided and then combined together after cross-attention layers so that the cross-attention layers can be tuned for better performance in concept erasing. The proposed method was evaluated for nudity, Van Gogh style, parachute object, church object removals. Self-generated images were also used for further performance improvement.

**Claims And Evidence:**

The claims in this work were partially supported by a number of experiments. However, there are also a number of issues regarding the claims such as "outperforming SOTA erasure approaches" due to lack of experiments, baselines, and so on. I will discuss these issues below. One claim that I would like to particularly discuss in here is on Section 4.1. There were two claimed limitations and were discussed with experiments. However, it is unclear if the analyses are sound and accurate.

- Limitation 1: it was well-known that diverse red-teaming attacks can generate erased concepts, but it is unclear if it was because of the gap between texts and images. It could come from the models' incapability since different models using the same text may or may not generate the same images. Thus, it will be hard to argue that the innate gap between text and image can be the reason of the vulnerability of the models.

- Limitation 2: There are two issues - this experiment may simply show the limitation of the ESD model itself or the limitation of the way of augmenting texts (which will be still finite words!). Thus, for the former, more models should be investigated to support this claim. For the latter, more sophisticated method such as using LLMs could be used. See the below recent work:
- Byung Hyun Lee et al., Concept Pinpoint Eraser for Text-to-image Diffusion Models via Residual Attention Gate, ICLR 2025.

While the overall idea of using images in training concept erasures, Figure 12 seems to show that more images adversely work against better retention performance for remaining concepts. This probably shows a clear disadvantage of the proposed method since images often have other concepts that should not be removed as well as the concept to remove. This could be more serious for multiple concept erasing (see the below work), thus, it seems important to verify the proposed method for these cases.
- Shilin Lu et al., MACE: Mass Concept Erasure in Diffusion Models, CVPR 2024.

**Essential References Not Discussed:**

See Masane Fuchi & Tomohiro Takagi, Erasing Concepts from Text-to-Image Diffusion Models with Few-shot Unlearning, BMVC 2024.

**Experimental Designs Or Analyses:**

This work should be evaluated for more datasets, baseline methods and more backbone networks. It is especially important to evaluate for the cases of having a series of concept erasing while retaining other concepts. See the below recent work:
- Shilin Lu et al., MACE: Mass Concept Erasure in Diffusion Models, CVPR 2024.

While Figure 9 looks nice, they are not useful enough to fully compare with other related works - usual tables with more detailed results will be needed. For example, see the Table 2 in the above work of (Lu et al., CVPR 2024), reporting the quantity of explicit content detected using the NudeNet detector on the I2P benchmark and the comparison of FID and CLIP on MS-COCO. Or see the Table 3 in that work on the assessment of Erasing 100 Artistic Styles. These are more informative over the current figure in this work. Thus, it seems important to demonstrate the capability of the proposed method using similar ways to these prior works + more recent works.

**Methods And Evaluation Criteria:**

The main idea was using images to aid text prompts for concept erasing. However, there were a couple of works regarding it. While the proposed method is different from them, it will be appropriate to properly discuss and compare with them. See the below recent work:
- Masane Fuchi & Tomohiro Takagi, Erasing Concepts from Text-to-Image Diffusion Models with Few-shot Unlearning, BMVC 2024.

**Other Comments Or Suggestions:**

Evaluating against red-team attacks seems important considering the limitation 1 in this work. See Ring-A-Bell (see below) or UnlearnDiff (Zhang et al., ECCV 2024) for more details.
- Yu-Lin Tsai et al., Ring-a-bell! how reliable are concept removal methods for diffusion models? ICLR 2024.

**Other Strengths And Weaknesses:**

It is unclear if this work will work for multiple concept erasures while retaining other remaining concepts. It seems important to discuss and demonstrate the scalability.

It is unclear if this work will work well for quite different images of the same concept. Since only one image could be used, it could provide a serious bias - any discussion for this?

**Questions For Authors:**

Could you explain why generated images were more effective over real images?

Was the sum of two latent vectors from text and image the best? Any other optimal way?

**Relation To Broader Scientific Literature:**

The idea of using images can also be similar to other machine unlearning works that focus on concept erasing since they are using both images and texts. It will be great if this work can discuss on those works and if possible, compare with some of them.

**Theoretical Claims:**

N/A

---

> ### Author Rebuttal · Authors · 2025-04-01
>
> Thank you for your valuable comments! Due to space constraints, we include `additional_tables.pdf` and figures at: https://anonymous.4open.science/r/icml25_rebuttal-8608. References to $\textcolor{blue}{\text{Table}}$ and $\textcolor{blue}{\text{Figure}}$ correspond to those provided **in this link**. A summary of all responses can be found at the end of the response to Reviewer VY4k.
>
> > Q1 Vulnerability from text-image gap or model incapability?
>
> **A1**: We believe the text-image gap **remains a key factor** to the vulnerability of diffusion models. Our reasoning is threefold:
>
> (1) **Scaling doesn’t close the gap**: We further conduct erasure on Stable Diffusion XL, a stronger model with improved text-image alignment. $\textcolor{blue}{\text{Figure 1}}$ shows similar vulnerabilities, indicating that the gap is ubiquitous regardless of scale.
>
> (2) **Attacks often rely on visual cues**. CCE[1] uses Textual Inversion from concept embeddings, and UnlearnDiff[2] optimizes prompts with target concept images, highlighting reliance on cross-modal cues.
>
> (3) **Prior work on modality misalignment**. Studies[3] support that modality gaps in CLIP (used in Stable Diffusion) impacts downstream tasks.
>
> [1]Circumventing Concept Erasure Methods For Text-to-Image Generative Models
>
> [2]To Generate or Not? Safety-Driven Unlearned Diffusion Models Are Still Easy to Generate Unsafe Images ... For Now
>
> [3]Mind the Gap: Understanding the Modality Gap in Multi-Modal Contrastive Representation Learning
>
> > Q2 Limitation 2 only shows ESD, and text augmentation is too simple.
>
> **A2**: We extend analysis to **SLD** (Safe Latent Diffusion) and **MACE** and apply LLM-based text augmentation[4]. $\textcolor{blue}{\text{Table 1}}$ shows **persistent text-based limitations** under more models and richer prompts.
>
> [4]Concept Pinpoint Eraser for Text-to-image Diffusion Models via Residual Attention Gate
>
> >  Q3 Potential disadvantages on multiple concept erasing task.
>
> **A3**: We agree multi-concept erasure is important. Since Co-Erasing is **plug-and-play**, we additionally integrate it with **MACE** and evaluate on multi-concept erasure. $\textcolor{blue}{\text{Table 2}}$ shows that Co-Erasing can complement MACE effectively.
>
> > Q4 Differences with other works involving images.
>
> **A4**: While other methods use images, **Co-Erasing differs in key ways**:
>
> (1) **Data Efficiency**: Unlike [5], which uses **external real images**, Co-Erasing uses **self-generated images**, reducing cost and aligning better with model's internal knowledge. See **A1** (uRE6) and **A2** (VY4K) for details.
>
> (2) **Preservation of Untargeted Concepts**:[5] does **not explicitly preserve** unrelated concepts, whereas Co-Erasing incorporates a text-guided refinement module to avoid unintended erasure.
>
> (3) **Modularity**: Co-Erasing is **plug-and-play** with methods like MACE and SLD, whereas [5] lacks flexibility.
>
> Additionally as shown in $\textcolor{blue}{\text{Table 3}}$, Co-Erasing **outperforms**[5] with a better trade-off.
>
> [5]Erasing Concepts from Text-to-Image Diffusion Models with Few-shot Unlearning
>
> > Q5 Potential bias when images of the same concept differ significantly and using only one image.
>
> **A5**: Co-Erasing is robust to visual diversity across images of the same concept:
>
> (1) We further analyze the distribution of image variations across concepts in $\textcolor{blue}{\text{Table 4}}$ and found **no strong relation** between visual diversity and performance drop, indicating Co-Erasing is **robust to image-level differences** within a concept.
>
> (2) At each training step, one image is randomly selected from the generated set (typically 50+), which mitigates overfitting to any single image and helps capture a common representation of the target.
>
> (3) **Text-guided refinement module** focuses on the semantic core of target concept, reducing impact from untargeted visual elements.
>
> > Q6 Evaluation against red-team attack: Ring-A-Bell.
>
> **A6**: We already evaluate against **UnlearnDiff** (UDA) in the paper. To further address this concern, we include **Ring-A-Bell** in $\textcolor{blue}{\text{Table 5}}$, which shows Co-Erasing can improve resistance against this attack.
>
> > Q7 The reason why generated images perform better.
>
> **A7**: Please see **A1** (uRE6) and **A2** (VY4K).
>
> > Q8 Other potential merging methods of text and image latent vectors.
>
> **A8**: Summation of text and image latent vectors follows **IP-Adapter**, widely adopted for integrating images into the generation process. To explore alternatives, we run experiments with different schedulers. $\textcolor{blue}{\text{Table 6}}$ shows slight performance gains. In future work we will further investigate optimal fusion designs.
>
> > Q9 More informative result comparison.
>
> **A9**: We show $\textcolor{blue}{\text{Table 7-15}}$, following MACE and AdvUnlearn[6], explicitly presenting quantitative results.
>
> [6]Defensive Unlearning with Adversarial Training for Robust Concept Erasure in Diffusion Models

---

### Decision · Program_Chairs · 2025-05-01

**Decision:**

Accept (poster)

**Comment:**

This paper introduces Co-Erasing, a novel framework for concept erasure in text-to-image diffusion models that aims to remove undesirable or harmful visual concepts (e.g., nudity, specific artistic styles, or objects) from generated outputs. The key innovation lies in its multimodal approach: combining both text prompts and image features, particularly self-generated images, to guide the concept erasure process more precisely than previous text-only methods.

Overall, the paper has clear motivation, innovative method, strong empirical results, and good writing, as agreed by most reviewers. Some concerns have been raised regarding the lack of experiments in some aspects, limited scope of concepts, additional computation costs. After author responses and discussions, most reviewers lean towards weak accept the paper. Thus, I would recommend 'Weak Accept'.